# Physics Informed Convex Artificial Neural Networks (PICANNs) for Optimal Transport based Density Estimation

## Abstract

Optimal Mass Transport (OMT) is a well-studied problem with a variety of applications in a diverse set of fields, ranging from Physics to Computer Vision and in particular Statistics and Data Science. Since the original formulation of Monge in 1781 significant theoretical progress been made on the existence, uniqueness and properties of the optimal transport maps. The actual numerical computation of the transport maps, particularly in high dimensions, remains a challenging problem. In the past decade several neural network based algorithms have been proposed to tackle this task. In this paper, building on recent developments of input convex neural networks and physics informed neural networks for solving PDE's, we propose a new Deep Learning approach to solve the continuous OMT problem. Our framework is based on Brenier's theorem, which reduces the continuous OMT problem to that of solving a nonlinear PDE of Monge-Ampere type whose solution is a convex function. To demonstrate the accuracy of our framework we compare our method to several other deep learning based algorithms. We then focus on applications to the ubiquitous density estimation and generative modeling tasks in statistics and machine learning. Finally as an example we present how our framework can be incorporated with an autoencoder to estimate an effective probabilistic generative model.

## 1 Introduction

Optimal Mass Transport (OMT) is a well-studied problem with a variety of applications in a diverse set of fields, ranging from physics to computer vision and in particular statistics and data science. In this paper we propose a new framework for the estimation of the solution to the $L^2$-optimal transport problem between two densities. Our algorithm, which is based on Brenier's theorem, and builds on recent developments of input convex neural networks and physics-informed neural networks for solving PDE's. Before we describe the contributions of the article in more detail, we will briefly summarize the motivation of our investigations and recent developments in the field.

**Density estimation and random sampling:** The density estimation problem is to estimate a smooth probability density based on a discrete finite set of observations. In traditional parametric density estimation techniques, the data is assumed to be drawn from a known parametric family of distributions. One of the most ubiquitous parametric techniques is Gaussian Mixture Modeling (McLachlan & Basford, 1988).

Nonparametric techniques were first proposed by Fix & Hodges (1951) (Silverman & Jones (1989)) to move away from rigid distributional assumptions. The most used approach is the kernel density estimation, which dates back to Rosenblatt (1952a) and Parzen (1962). Many challenges remain regarding the implementation and practical performance of kernel density estimators, including in particular, the bandwidth selection and the lack of local adaptivity resulting in a large sensitivity to outliers (Loader et al., 1999). These problems are particularly exacerbated in high dimensions with the curse of dimensionality.

Recently, diffeomorphic transformation-based algorithms have been proposed to tackle this problem (Dinh et al., 2017; Marzouk et al., 2016; Younes, 2020; Bauer et al., 2017). The basic concept of transformation-based algorithms is to find a diffeomorphic mapping between a reference proba-

bility distribution and the unknown target distribution, from which the data is drawn. Consequently, transformation-based density estimation leads at the same time to an efficient generative model, as new samples from the estimated density can be generated at a low cost by sampling from the reference density and transforming the samples by the estimated transformation. The fundamental problem in diffeomorphic transformation-based approaches is how to estimate and select the transformation: from a theoretical point of view there exists an infinite set of transformations that map two given probability densities onto each other. Recently, several deep learning methods have been devised for this task, where Normalizing Flows (NF) stand out among these methods. Examples of such models include Real NVP (Dinh et al., 2017), Masked Autoregressive Flows (Papamakarios et al., 2017), iResNets (Behrmann et al., 2019), Flow++ (Ho et al., 2019) and Glow (Kingma & Dhariwal, 2018). For a review of the vast NF literature, we refer to the the overview article (Kobyzev et al., 2020). Although these methods have shown to perform well in density estimation applications, the interpretability of the obtained transformation is less clear, e.g. in Real NVP (Dinh et al., 2017), the solution selection is obtained by restricitng the transformations to the class of diffeomorphisms with triangular Jacobians that are easy to invert, which is closely related to the Knothe-Rosenblatt rearrangement (Knothe, 1957; Rosenblatt, 1952b).

**Optimal mass transport:** Optimal mass transport, on the other hand, formulates the transport map selection as the minimizer of a cost function (Villani, 2008; 2003). The optimal transportation cost induces a metric structure, the Wasserstein metric, on the space of probability densities and is sometimes referred to as the Earth Mover's Distance. This theory, which dates back to 1781, was originally formulated by the French mathematician Gaspard Monge (1781). The difficulty in applying this framework to the proposed density estimation problem lies in solving the corresponding optimization problem, which in a dimension greater than one is highly non trivial. The fully discrete OMT problem (optimal assignment problem) can be solved using linear programming and can be approximated by the Sinkhorn algorithm (Cuturi, 2013a; Papadakis, 2015). However, these algorithms do not lead to a continuous transformation map and thus can't be used for the proposed diffeomorphic density estimation and generative modelling. Previous algorithmic solutions for the continuous OMT problem include fluid mechanics-based approaches (Benamou & Brenier, 2000), finite element or finite difference-based methods (Benamou et al., 2010; Benamou & Duval, 2019) and steepest descent-based energy minimization approaches (Angenent et al., 2003; Carlier et al., 2010; Loeper & Rapetti, 2005).

In recent years, several deep learning methods have been deployed for solving the OMT problem. In these methods, the OMT problem is typically embedded in the loss function for the neural network model. Recent work by Makkuva et al. (2020) proposed to approximate the OMT map as the solution of min-max optimization using input convex neural networks (ICNN), see Amos et al. (2017). The min-max nature of this algorithm arises from the need to train an ICNN to represent a convex function and the conjugate of the convex function. Building upon this approach, Korotin et al. (2019) imposed a cyclic regularisation that converts the min-max optimization problem to a standard minimization problem. This change results in a faster converging algorithm that scales well to higher dimensions and also prevents convergence to local saddle points and instabilities during training, as is the case in the min-max algorithm.

Another class of neural networks which have been proposed to solve OMT problems are Generative Adversarial Networks(GANs) (Goodfellow et al., 2014). GANs are defined through a min-max game of two neural networks where one of the networks tries to generate new samples from a data distribution, while the other network judges whether these generated samples originate from the data population or not. Later, Gulrajani et al. (2017) proposed using the Wasserstein-1 distance in GANs instead of the Jensen-Shannon divergence between the generated distribution and the data distribution as in the original formulation. They demonstrated that this new loss functions leads to better stability of the training of networks attributed to the Wasserstein metric being well defined even when the two distributions do not share the same support.

**Contributions:** In this paper, we propose a different deep learning-based framework to approximate the optimal transport maps. The approach we present relies on Brenier's celebrated theorem (Brenier, 1991), thereby reducing the optimal transport problem to that of solving a partial differential equation: a Monge-Ampere type equation. We frame this PDE in the recently developed paradigm of Physics Informed Neural Networks (PINNs) (Raissi et al., 2017). Similar to other deep learning-based algorithms, our framework directly inherits the dimensional scalability of neural networks (Shin et al., 2020), which traditional finite element or finite difference methods for solving

PDEs do not possess. Brenier's theorem further states that the optimal transport map is given by the gradient of a convex function- the Brenier potential. To incorporate this information in our PINN approach, we parameterize the Brenier potential using an ICNN, thereby guaranteeing its convexity.

We test the accuracy of our OMT solver on numerous synthetic examples for which analytical solutions are known. Our experiments show that our algorithm indeed approximates the true solution well, even in high dimensions. To further quantify the performance of the new framework, we compare it to two other deep learning-based algorithms, for which we guided the selection by the results of the recent benchmarking paper by Korotin et al. (2021), in which they evaluate the methods presented in Seguy et al. (2017); Nhan Dam et al. (2019); Taghvaei & Jalali (2019); Makkuva et al. (2020); Liu et al. (2019); Mallasto et al. (2019); Korotin et al. (2019). We restricted our comparision to the algorithms of Makkuva et al. (2020) and Korotin et al. (2019), as these two showed the best performance in this benchmark. Our results showed that the newly proposed method significantly outperforms these methods in terms of accuracy.

As an explicit application of our solution of OMT, we focus on the density estimation problem. In synthetic examples, we show that we can estimate the true density based on a limited amount of samples. We compare the results of the proposed method to four other density estimation algorithm: the two OMT based algorithms mentioned above, and two methods from the family of normalizing flows: RealNVP (Dinh et al. (2016)) and iResNet (Behrmann et al. (2019)). Finally we demonstrate how our framework can be combined with a traditional autoencoder to obtain a generative framework. In accordance with the best practices for reproducible research, we are providing an open-source version of the code, which is publicly available on github.

## 2 OMT USING DEEP LEARNING

In this section, we will present our framework for solving the Optimal Mass Transport (OMT) problem. Our approach will combine methods of deep learning with the celebrated theorem of Brenier, which reduces the solution of the OMT problem to solving a Monge-Ampere type equation. To be more precise, we will tackle this problem by embedding the Monge-Ampere equation into the broadly applicable concept of Physics Informed Neural Networks.

### 2.1 MATHEMATICAL BACKGROUND OF OMT

We start by summarizing the mathematical background of OMT, including a description of Brenier's theorem. For more information we refer to the vast literature on OMT, see e.g., Villani (2003; 2008). Let $\Omega$ be a convex and bounded domain of $\mathbb{R}^n$ and let $dx$ denote the standard measure on $\mathbb{R}^n$. For simplicity, we restrict our presentation to the set $\mathcal{P}(\Omega)$ of all absolutely continuous measures on $\Omega$, i.e., $\mathcal{P}(\Omega) \ni \mu = f dx$ with $f \in L^1(\Omega)$, such that $\int_\Omega f dx = 1$. From here, on we will identify the measure $\mu$ with its density function $f$. We aim to minimize the cost of transporting a density $\mu$ to a density $\nu$ using a (transport) map $T$, which leads to the so-called Monge Optimal Transport Problem. We will consider only the special case of a quadratic cost function as Brenier's theorem, which forms the basis of our algorithm, is only true for this type of cost function.

**Definition 2.1 ($L^2$-Monge Optimal Transport Problem)** *Given $\mu, \nu \in \mathcal{P}(\Omega)$, minimize $\mathbb{M}(T) = \int_\Omega \|x - T(x)\|^2 d\mu(x)$ over all $\mu$-measureable maps $T : \Omega \to \Omega$ subject to $\nu = T_*\mu$. We will call an optimal $T$ an optimal transport map.*

Here, the constraint is formulated in terms of the push forward action of a measurable map $T : \Omega \to \Omega$, which is defined via $T_*\mu(B) = \mu(T^{-1}(B))$, for every measurable set $A \subset \Omega$. By a change of coordinates, the constraint $T_*\mu = T_*(f dx) = \nu = g dx$ can be thus reduced to the equation

$$f(x) = g(T(x))|\det(DT(x))|. \tag{1}$$

This equation can also be expressed via the pullback action as $\mu = T^*\nu := (T^{-1})_*\nu$. The existence and uniqueness of an optimal transport map is not guaranteed. We will see, that in our situation, i.e., for absolutely continuous measures, the existence and uniqueness is indeed guaranteed. First, we will introduce a more general formulation of the Monge problem, called Kantorovich formulation. Therefore, we define the space of all transport plans $\Pi(\mu, \nu)$, i.e., of all measures on the product space $\Omega \times \Omega$, such that the first marginal is $\mu$ and the second marginal is $\nu$. Then we have:

**Definition 2.2 ($L^2$-Kantorovich's Optimal Transport Problem)** *Given $\mu, \nu \in \mathcal{P}(\Omega)$, minimize $\mathbb{K}(\pi) = \int_{\Omega \times \Omega} \|x - y\|^2 d\pi(x, y)$ over all $\pi \in \Pi(\mu, \nu)$.*

Note that the $L^2$-Wasserstein metric $W_2(\mu, \nu)$ between $\mu$ and $\nu$ is defined as the infimum of $\mathbb{K}$. We will now formulate Brenier's theorem, which guarantees the existence of an optimal transport map and will be the central building block of our algorithm:

**Theorem 2.3 (Brenier (1991))** *Let $\mu, \nu \in \mathcal{P}(\Omega)$. Then there exists a unique optimal transport plan $\pi^* \in \Pi(f, g)$, which is given by $\pi^*(x, y) = (\mathrm{id} \times T)$ where $T = \nabla u$ is the gradient of a convex function $u$ that pushes $\mu$ forward to $\nu$, i.e., $(\nabla u)_* \mu = \nu$. The inverse $T^{-1}$ is also given by the gradient of a convex function that is the Legendre transform of the convex function $u$.*

Thus, Brenier's Theorem guarantees the existence and the uniqueness of the optimal transport map of the OMT problem. Consequently, we can determine this optimal transport map by solving for the function $u$ in the form of a Monge-Ampère equation:

$$\det(D^2(u)(x)) \cdot g(\nabla u(x)) = f(x) \tag{2}$$

where $D^2$ is the Hessian, $\mu = f dx$ and $\nu = g dx$. We obtain equation 2 directly from equation 1 using the constraint that $T = \nabla u$ as required by Brenier's theorem. We will also refer to this map as the Brenier map. This map is a diffeomorphism as it is a gradient of a strictly convex function. In general, the **Monge–Ampère equation** is a nonlinear second-order partial differential equation. If we limit ourselves to convex solutions, then the differential equation is elliptic. If the two densities $f$ and $g$ are smooth ($C^\infty$), positive and absolutely continuous with respect to each other then an unique smooth convex solution is guaranteed to exist. See e.g. Forzani & Maldonado (2004); Bakelman (1983); De Philippis & Figalli (2013) for further details on existence and regularity of solutions to this equation. Using methods of classical numerical analysis, Brenier's theorem has been used e.g. in Peyré et al. (2019) to obtain a numerical framework for the continous OMT problem. In the following section we will propose a new discretization to this problem, which will make use of recent advances in deep learning.

## 2.2 SOLVING OMT USING PINNS

Physics Informed Neural Networks (PINNs) were proposed by Raissi et al. (2017) to solve general nonlinear partial differential equations (PDEs). The basic concept is to use the universal approximation property of deep neural networks to represent the solution of a PDE via a network. Using the automatic differentiation capability of modern machine learning frameworks, a loss function is formulated, such that its minimizer solves the PDE in a weak sense. Such a loss function encodes the structured information, which results in the amplification of the information content of the data the network sees (Raissi et al., 2017). This formulation of the PDE results in good generalization even when only few training examples are available. PINNs have found widespread applications in a short period of time since their introduction. These applications include a wide variety of PDEs, including the Navier-Stokes equation (Jin et al., 2021), nonlinear stochastic PDEs (Zhang et al., 2020) or Allen Cahn PDEs (McClenny & Braga-Neto, 2020).

In this work, we propose to use the PINN approach to solve the Monge-Ampere equation, as presented in equation 2, and hence implicitly the OMT problem. This equation has been extensively studied and the properties of its solutions are well established. By Theorem 2.3, we know that the solution is given by a convex function $u$. Recently, Amos et al. (2017) proposed a new architecture of neural networks, Input Convex Neural Networks (ICNNs), that explicitly constrains the function approximated by the network to be convex. Furthermore, ICNNs inherit the universal approximation powers of feedforward networks (Amos et al., 2017, Proposition 2) and thus we can approximate any convex function arbitrarily well by using an ICNN architecture of sufficient depth and width. Consequently, this architecture naturally lends itself to our proposed application, as it directly encodes Brenier's theorem. In the ICNN architecture, the activation function is a nondecreasing convex function and the internal weights ($W_n^{(x)}$) are constrained to be non-negative; see Figure 4 in Appendix A for a schematic description of this class of networks. This architecture is derived from two simple facts: non-negative sums of convex functions are also convex, and the composition of a convex and convex nondecreasing function is again convex.

We are now ready to describe our procedure for solving the continuous OMT problem: given absolutely continuous densities $\mu = f dx$ and $\nu = g dx$ we reduce, using Brenier's theorem, the OMT

problem to the Monge-Ampere equation 2. Using the PINN approach to solve this PDE leads to consider the loss function

$$\| \det(D^2(u)) \cdot g(\nabla u) - f \|_{L^2}^2,$$ (3)

which corresponds to a weak formulation of the PDE. To incorporate the convexity of $u$ we construct the solution space using an ICNN of sufficient depth and width. This loss function scales exponentially with respect to dimensions of the domain of integration $\Omega$. This stems from the fact that the Monge-Ampere equation contains the determinant of the Jacobian. Using the positive definiteness of the Jacobian (being the Hessian of a convex function) one could potentially make use of efficient determinant estimators, such as Stochastic Chebyshev Expansions as developed in Han et al. (2018), which would scales linearly in dimension. Once we have estimated the optimal transport map, the $L^2$-Wasserstein metric between $\mu$ and $\nu$ is given by

$$\int \| x - \nabla u(x) \|^2 \, g(x) dx.$$ (4)

We call this combination of the PINN approach with the ICNN structure, Physics Informed Convex Artificial Neural Networks (PICANNs).

In several applications, we are interested in computing the inverse transformation at the same time. By a duality argument, we know that this map is also given by the gradient of a convex function. Thus, we use a second ICNN (with the same architecture) to compute the inverse optimal transport map ($\nabla v$) by solving the minimization problem:

$$\| \nabla v(\nabla u(x)) - x \|_{L^2},$$ (5)

where $\nabla u$ is the optimal transport map solving $(\nabla u)_* \mu = \nu$, c.f. Figure 4 in Appendix A.

## 2.3 DIFFEOMORPHIC RANDOM SAMPLING AND DENSITY ESTIMATION

In many applications, such as the Bayesian estimation, we can evaluate the density rather easily but generating samples from a given density is not trivial. Traditional methods include Markov Chain Monte Carlo methods, e.g., the Metropolis Hastings algorithm (Hastings, 1970). An alternative idea is to use diffeomorphic density matching between the given density $\nu$ and a standard density $\mu$ from which samples can be drawn easily. Once we have calculated the transport map, standard samples are transformed by the push-forward diffeomorphism to generate samples from the target density $\nu$. This approach has been followed in several articles, where the optimal transport map selection was based on both, the Fisher-Rao metric (Bauer et al., 2017) and the Knothe–Rosenblatt rearrangement (Marzouk et al., 2016). The efficient implementation of the present paper directly leads to an efficient random sampling algorithm in high dimensions.

We now recall the density estimation problem using the OMT framework. We are given samples $x_i$ drawn from an unknown density $\mu \in \mathcal{P}(\Omega)$ that we aim to estimate. The main idea of our algorithm is to represent the unknown density as the pullback via a (diffeomorphic) Brenier map $\nabla u$ of a given background density $\nu = g dx$, i.e., $(\nabla u)^* \nu = \mu$ or equivalently using the push forward action as $(\nabla u)_* \mu = \nu$, where the pull back and push forward of a density are defined in Section 2.1.

As we do not have an explicit target density, but only a finite number of samples, we need to find a replacement for the $L^2$-norm used in equation 3 to estimate the transport map $\nabla u$. We do this by maximizing the log-likelihood of the data with respect to the density $(\nabla u)^* \nu$:

$$\frac{1}{N} \sum_i \log \left( \det(D^2(u(x_i))) \cdot g(\nabla u(x_i)) \right).$$ (6)

Using our PINNs framework, we represent the convex function $u$ again via an ICNN of the same architecture , which serves as an implicit regularizer, see Sivaprasad et al. (2021). This equation can be alternatively interpreted as minimizing the empirical Kullback-Leibler divergence between $\mu$ and the pullback of the background density $\nu$. To generate new samples from the estimated density, we use the inverse map to transform the samples from the background density $\nu$. We calculate the inverse map using a second neural network and explicit loss function given by equation 5.

# 3 EXPERIMENTAL RESULTS

In this section, we will detail our implementation and present several experiments demonstrating both the applicability and accuracy of our framework. In particular, we will compare our results in several experiments to state-of-the-art deep learning-based OMT solvers, and we will show that we outperform these methods in terms of accuracy.

## 3.1 NETWORK DETAILS

As explained in Section 2.2, we use an ICNN architecture for both the forward and the backward map in all of our experiments, c.f. Figure 4 in Appendix A. As with every deep learning approach, we need to tune the hyperparameters, including width/depth of the network, activation functions and batch size. The width of the network needs to increase with the dimension of the ambient space of the data to ensure sufficient flexibility. For our experiments in lower dimensions, we used a network with three hidden layers with 128 neurons in each layer, whereas for experiments in 30d, we used a network with four hidden layers with 128 neurons in each layer. To initialize the network, we first train the networks to learn the identity transformation, i.e., $\nabla u = I$, which we use as the initial starting point for all our experiments. In all our experiments, 10,000 target samples were used.

To guarantee the convexity of the output function, the activation functions need to be convex and non-decreasing. Since simple ReLUs are not strictly convex and have a vanishing second derivative almost everywhere, we experimented with the family of Rectified Power Units (RePUs), the log exponential family and the 'Softplus' function. The Softplus function to the power of $\alpha$, which is defined via $\text{Softplus}^\alpha(x) = (\log(1 + \exp x))^\alpha$, turned out to be best suited for our applications, where we chose $\alpha = 1.1$. In particular our experiments suggested that networks with this activation function were able to generalize well to regions where no or only limited training data were available.

## 3.2 VALIDATION AND COMPARISON TO OTHER METHODS

To demonstrate the accuracy of our implementation, we present two different types of experiments: first we will conduct a series of experiments in which analytic solutions to the OMT problem are available, where we compare our results to results obtained with two state-of-the-art deep learning-based OMT solvers: OT-ICNN from Makkuva et al. (2020) and W2Gen from Korotin et al. (2019). We choose these two specific algorithms among the available plethora of available OMT solvers based on the recent benchmark paper by Korotin et al. (2021). In a second set of experiments we will consider density estimation problems, where we will compare the quality of our results with four other methods; the two OMT solvers from above and two methods from the family of normalizing flows: RealNVP as introduced in Dinh et al. (2016) and iResNet from Behrmann et al. (2019).
**Approximating the OMT map:** Since both OT-ICNN and W2Gen are also based on an ICNN structure, we were able to choose the same architecture with same hyperparameters for all three algorithms, thereby ensuring a fair comparison. Nevertheless, we want to emphasize that these parameters could be further fine tuned for all the algorithms and specific experiments to improve the results. Nevertheless, with exception of several experiments for the OT-ICNN solver, we observed a good convergence behavior. We present selected convergence graphs in appendix B. Furthermore we demonstrate the scalabilty of our algorithm by performing the same experiment in dimensions 2, 3, 5, 8, 15 and 30.

We do not present comparisons of our approach to more traditional OMT algorithms such as the Sinkhorn algorithm (Cuturi, 2013b) or the linear programming approaches (Peyré et al., 2019), as these frameworks, although they approximate the OMT distances, do not compute the continuous optimal transport map, which is essential for the proposed density estimation. While finite element or finite difference based Monge-Ampere solvers, see e.g. (Benamou & Duval, 2019; Jacobs & Léger, 2020; Benamou & Brenier, 2000), calculate the continuous OMT map, they are not suitable in dimensions greater than two or three.

To quantify the quality of an estimated transport plan $T$, we calculate the $\mathcal{L}^2$-UVP unexplained variance percentage (UVP), which is given by $\mathcal{L}^2\text{-UVP}(T) = 100 \cdot \|T - T^*\|_{\mathcal{L}^2(\mu)}^2 / \text{Var}(\nu)$, where $T^*$ denotes the (analytic) optimal transport plan. In appendix D we present in addition the percentage error between the analytic Wasserstein distance and the approximated distance, which can be viewed as second, albeit coarser, measure of quality. Our first series of experiments is the same as

| $\mathcal{L}^2$- **Unexplained Variance Percentage ($\mathcal{L}^2$-UVP)** | | | | | | | |
|---|---|---|---|---|---|---|---|
| **Experiment** | **Method** | **Dimensions** | | | | | |
| | | **2d** | **3d** | **5d** | **8d** | **15d** | **30d** |
| Random Cvx Function | PICANNs | **0.004** | **0.007** | **0.021** | **0.034** | **0.144** | **0.38** |
| | OT-ICNN | 0.043 | 0.052 | 0.145 | 0.276 | 0.746 | 3.98 |
| | W2GEN | 0.040 | 0.043 | 0.046 | 0.052 | 0.150 | 0.60 |
| Random Gaussian | PICANNs | 0.33 | 0.15 | **0.15** | 0.28 | **0.30** | 1.13 |
| | OT-ICNN | 0.28 | 0.71 | 0.86 | 2.38 | 2.84 | 2.24 |
| | W2GEN | **0.17** | **0.14** | 0.16 | **0.23** | 0.37 | **0.67** |
| Annulus | PICANNs | **0.29** | **0.43** | **0.63** | **1.61** | **7.53** | **21.71** |
| | OT-ICNN | 23.84 | 9.98 | 28.21 | 43.87 | 44.52 | 2725.15 |
| | W2GEN | 1.33 | 6.86 | 18.31 | 20.50 | 23.28 | 34.19 |

Table 1: We present a comparison between our PICANN approach and OT-ICNN from Makkuva et al. (2020) and W2Gen from Korotin et al. (2019). We present the L2-UVP for three experiments. In all these experiments, the source density is the unit Gaussian. The target density in the case of the "Random Cvx Function" experiment is the unit Gaussian deformed by the gradient of a random convex function. In the case "Random Gaussian", the target density is another Gaussian with a randomly sampled mean and co-variance matrix. In the third experiment, the target density is the annulus distribution. The results in the first two experiments are averages of 20 realizations.

in the benchmark paper (Korotin et al., 2021): we use the gradient of a random convex function to transport the unit Gaussian to a random density. By Brenier's Theorem, as the optimal transport map is the gradient of a convex function, this map is the optimal transport map. In each dimension, we repeated this experiment 20 times to compute the error statistics, which are presented in Table 1. Wheras all three algorithms seem to work well for this experiment, PICANNs consistently outperform the other algorithms. As already observed in Korotin et al. (2021), this experiment is favoring the ICNN architecture, as the true solution was chosen to be of the same nature, which explains the nearly perfect performance of all three algorithms. Next, we turn to cases where the analytical solution is known in closed form. The first is the special case where both densities are from a family of Gaussians distributions. In that case, the OMT map is simply given by an affine transform and the OMT distance is again given in closed form. We again repeat this experiment 20 times for each dimension, where we generate Gaussian distributions with a random mean and covariances. Here the means are sampled from a uniform distribution on $[-1, 1]$. To construct the random covariance matrices, we recall that we need to enforce the matrix to be positive definite and symmetric. Therefore, we generate a random matrix $A$ of dimension $d \times 3d$, where $d$ is the dimension of the space and where the entries are i.i.d. chosen from a uniform distribution on $[0, 0.75]$. Then, a random covariance matrix can be constructed by letting $\Sigma = AA^T$ (the particular form of $\Sigma$ almost surely guarantees positive definiteness). The results and comparisons with the other two methods are again presented in Table 1. In general, all three algorithms still lead to a good approximation, where one can see that W2Gen and PICANNs are performing significantly better than OT-ICNN. To further validate our algorithm, we choose a more challenging problem: an annulus density for which we know the transport map in closed form. The annulus distribution is given by a push forward of the Gaussian distribution by a gradient of a radially symmetric convex function and is given by $f = g((X^T X)X) \cdot 3(X^T X)^d$ where $g$ is the unit Gaussian and $d$ is the number of dimensions. The transport map $X \mapsto (X^T X)X$ is the gradient of the convex function $\frac{1}{4}(X^T X)^2$ and thus by Brenier's theorem the optimal transport map. The results in Table 1 clearly demonstrates that, in particular for this more challenging problem, the proposed PICANNs outperform the other algorithms by orders of magnitudes.

**Density estimation comparisons:** Next we consider the problem of estimating a continuous density from discrete finite samples. In this part we will compare our method in addition to two methods from the normalized flow family, namely RealNVP and iResNet. For both these methods we choose the standard network structure as used in the examples provided by the developers, i.e., for RealNVP we used a fully connected network with 3 hidden layers with 256 neurons in each layer and for iResNets we used a fully connected network with 4 hidden layers with 128 neurons in each layer.

We want to emphasize that NODE and normalized flows do not solve the OMT problem and consequently there is no physically motivated optimality conditions of the transport map generated by these methods. This is also reflected in our experiments: in the OMT approach we search for the map that is as close as possible to the identity among all transformations that satisfy the required density matching constraint (note, that there is an infinite dimensional set of transformations that satisfy that satisfy this constraints). Consequently the obtained transformation maps for OMT are more regular as compared to the maps obtained with other density estimation methods such as iResNet or RealNVP, cf. Figures 1,2,7 and 8.

As a first example, we consider again the annulus density, that we introduced in the last experiment of the previous section. Figure 1 shows the comparison of five different methods for estimating the density as well as the transport maps. To quantify the ability to approximate the true density we also report the KL-divergence between the estimated density (i.e., the push-forward of the unit gaussian by the estimated transport map) and the given samples. While W2Gen, RealNVP and iResNet all showed a decent estimation of the annulus density, the quality of their approximations is clearly inferior to our PICANNs approach. Even after extensive parameter tuning we could not get the OT-ICNN algorithm to converge to the true density. Finally, we note that RealNVP and iResNet do not aim to compute the $L^2$ optimal transport map. This is exemplified in Figure 1, where one can see that the estimated maps are very far from the optimal transport map for these two algorithms. A second example in which we consider a nonsymmetric distribution that has been constructed in Bauer et al. (2017) can be seen in Figure 2. Similar as for the annulus density our PICANNs approach produces the best result. Further examples and comparisons can be found in Appendix C.

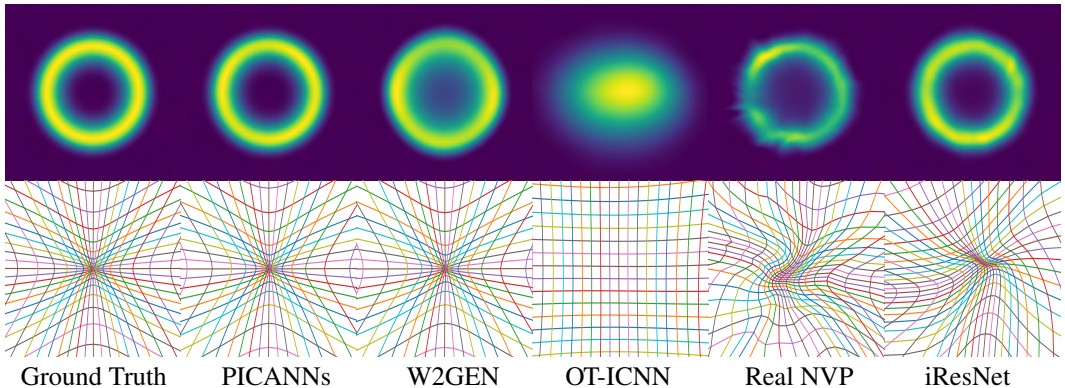

| Ground Truth | PICANNs | W2GEN | OT-ICNN | Real NVP | iResNet |

Figure 1: Comparison of different density estimation frameworks. First line: the estimated densities. Second line: the corresponding transport map.

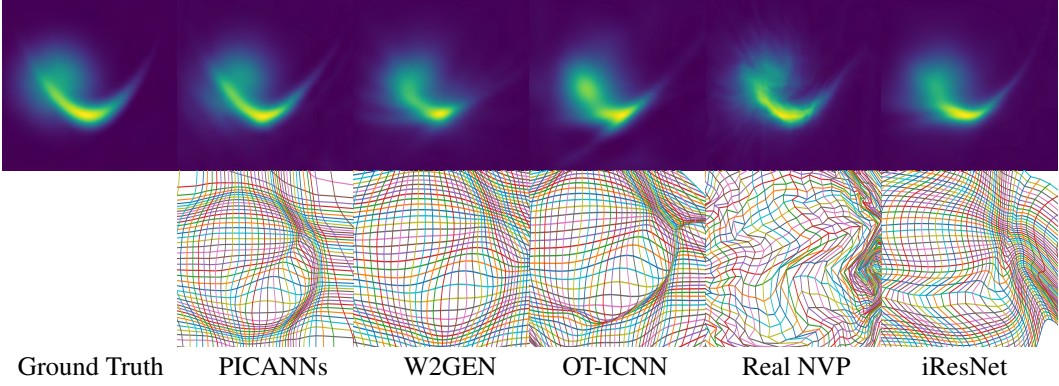

| Ground Truth | PICANNs | W2GEN | OT-ICNN | Real NVP | iResNet |

Figure 2: Comparison of different density estimation frameworks. First line: the estimated densities. Second line: the corresponding transport map. Note, that we do not have access to the analytic optimal transport map in this example.

### 3.3 GENERATIVE MODELING WITH PICANNS

Finally, we present preliminary results to demonstrate the application of the PICANN approach to develop a generative model. It has been well established that for most real world image modeling applications the support of the target probability distribution is a lower dimensional manifold Gerber et al. (2009); Lee & Verleysen (2007). Autoencoders are very effective in non-linear dimensionality reduction and map high-dimensional data to a latent space of lower dimensions, which then can be transformed back to the original space using the "decoder" part of the network. One of the major shortcoming of an autoencoder is that the distribution of data mapped in to the latent space is not know and cannot be assumed to be Gaussian. In Appendix A, Fig 5 we summarizes how the autoencoder can be naturally included in our PICANN approach to obtain an efficient generative model for high-dimensional data.

To demonstrate the generative algorithm in a toy example, we consider the MNIST dataset encoded using a simple fully connected autoencoder to a latent space of dimension 2. Figure 3 shows the encoded data points with each class assigned a unique color. We train a PICANN network of 5 hidden layers with 128 neurons in each layer, to learn the forward and the inverse mapping from a Gaussian with the mean and covariance of the encoded data to the unknown "latent MNIST distribution". Figure 3 presents comparisons to a similar setup with the four other methods discussed in the previous section. The results show significant differences in the quality of the generated samples, where PICANNs and RealNVP visually lead to the best results. In future work we plan to continue these investigations, by applying the setup to more challenging image sets and carefully comparing the results using state-of-the-art quality metrics.

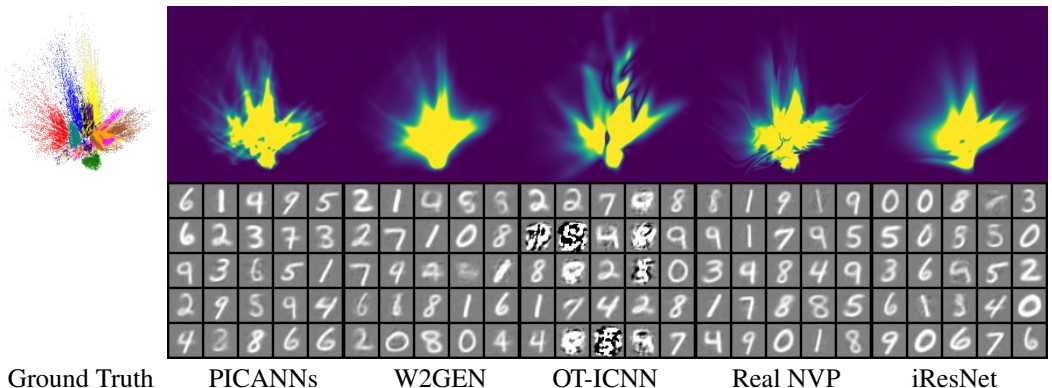

| Ground Truth | PICANNs | W2GEN | OT-ICNN | Real NVP | iResNet |

Figure 3: Generative model: this figure details the application of our generative framework to the MNIST dataset. Five different algorithms were trained to estimate this distribution and learn the forward and inverse transport map between the encoded 'MNIST Distribution' and a Gaussian. The estimated density can be seen in the first line. The second line shows 25 samples passed through the decoder part of the autoencoder for each of the methods.

## 4 CONCLUSION

In this paper, we use the $L^2$-Wasserstein metric and optimal mass transport (OMT) theory to formulate a density estimation and generative modeling framework. We develop a new deep learning-based solver for the continuous OMT problem, which is rooted in Brenier's celebrated theorem. This theorem allows us to formulate the density estimation problem as a solution to a nonlinear PDE – a Monge-Ampere equation. Recent developments in deep learning for PDEs, namely PINNS and ICNNs, allow us to develop an efficient solver. We demonstrate the accuracy of our framework by comparing our results to analytic Wasserstein distances. To further quantify the quality of our results we compare them to the results obtained with the two best performing algorithms of the recent benchmark paper for deep learning based OMT solvers (Korotin et al., 2021). Our experiments show that our approach significantly outperforms these methods in term of accuracy. Finally, we present examples of diffeomorphic density estimation within our framework and showcase an example of a generative model.

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

# A   NETWORK STRUCTURES

In this section we present schematics of our PICANNs approach (Figure 4) and how it can be included in a generative model (Figure 5).

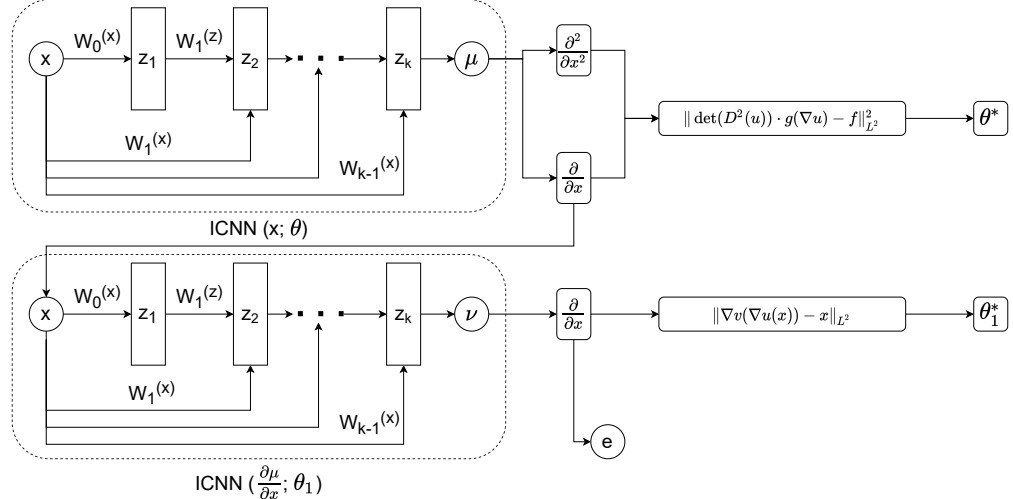

Figure 4: PICANN architecture. We present how a combination of two ICNN networks can be used to learn the forward and the inverse map between two distributions. Both these networks are trained independently with their respective loss functions. The inverse network uses the gradient of the output of the first network as its input.

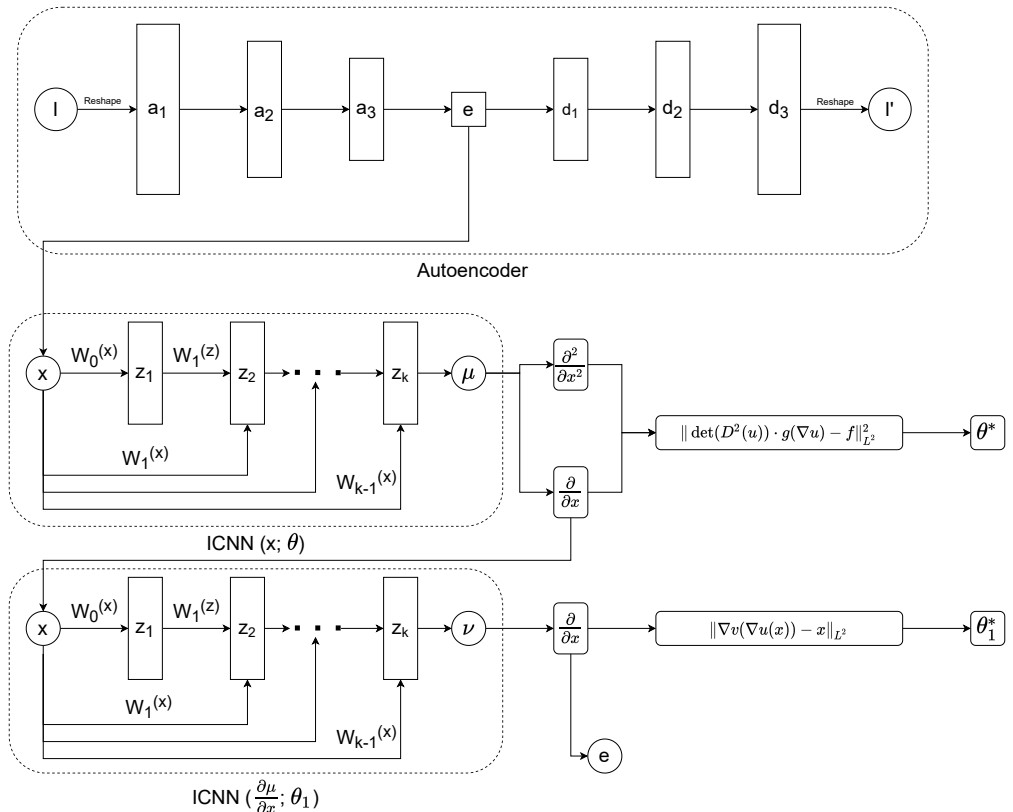

Figure 5: How a combination of an autoencoder with our PICANN approach can be used to develop a generative model. Note how the latent space of the autoencoder becomes the input to the PICANN network. In such a setting, the PICANN estimates the latent space density and samples from the estimated distribution. Using random samples from this distribution, one can pass them through the decoder to generate new samples.

## B CONVERGENCE OF EXPERIMENTS

In this appendix (Figure 6) we present the convergence plots for a several experiments of Table 1. It should be noted that PICANNs and W2GEN are pure minimisation algorithms while OT-ICNN is a min-max algorithm and hence the covergence plots for OT-ICNN differ from the convergence plots for the other two algorithms. These convergence plots show that PICANNs and W2GEN converge for all the experiments presented here while the OT-ICNN, as remarked in Section 3.2, does not converge in the case of the annulus experiments. This is also the case for the annulus experiments in higher dimensions (not shown here). In table 2 we also present the average timing for 100 Epochs for these three algorithms in different dimensions.

| Method | 2d | 3d | 5d | 8d | 15d | 30d |
|---|---|---|---|---|---|---|
| PICANNs | 90s | 95s | 130s | 165s | 233s | 310s |
| W2Gen | 110s | 110s | 112s | 115s | 118s | 120s |
| OT-ICNN | 180s | 180s | 185s | 186s | 197s | 210s |

Table 2: Computational cost for 100 epochs in various dimensions.

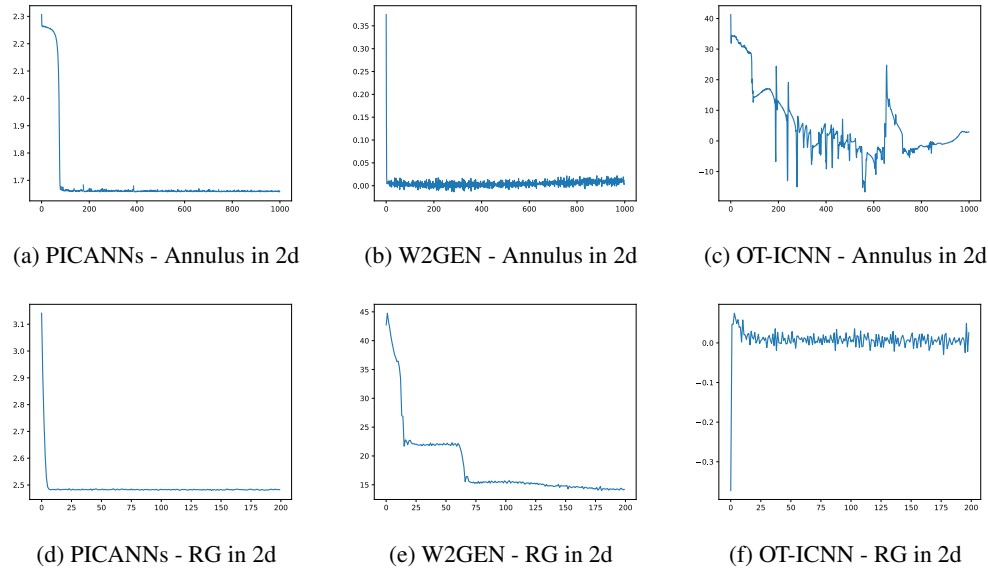

(a) PICANNs - Annulus in 2d     (b) W2GEN - Annulus in 2d     (c) OT-ICNN - Annulus in 2d

(d) PICANNs - RG in 2d     (e) W2GEN - RG in 2d     (f) OT-ICNN - RG in 2d

Figure 6: In this figure we present the convergence plots for PICANNs, W2GEN and OT-ICNN. In row 1 we present convergence plots for the annulus experiment in 2d for PICANNs, W2GEN and OT-ICNN in Panels (a), (b) and (c) respectively. In row 2 we present convergence plots for the gaussian experiments performed using again PICANNs, W2GEN and OT-ICNN in Panels (d), (e) and (f) respectively. As the OT-ICNN algorithm is a min-max algorithm these convergence plots have to be read slightly differently. For the gaussian experiment, the loss starts with small negative values and climbs up to a small positive quantity but around the 50 epoch mark it has converged to about 0 and just bounces in that neighbourhood thereafter. The OT-ICNN algorithm does not converge for the annulus experiment, as remarked in Section 3.2.

## C    FURTHER DENSITY ESTIMATION EXAMPLES AND COMPARISONS

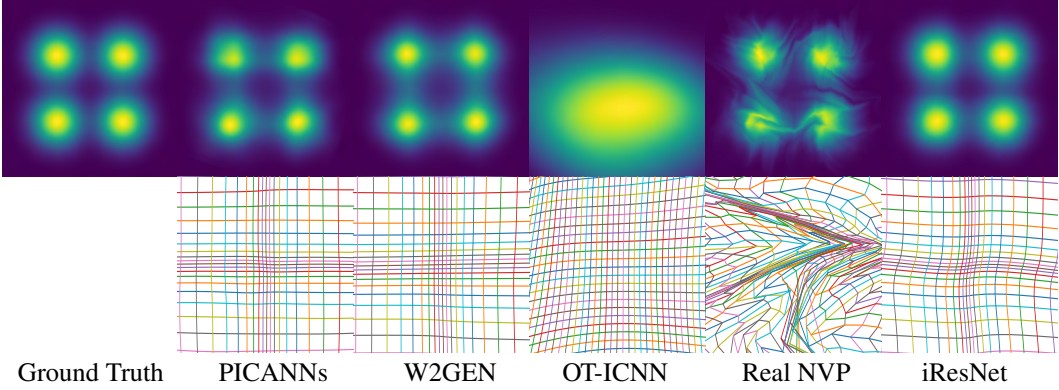

Ground Truth     PICANNs     W2GEN     OT-ICNN     Real NVP     iResNet

Figure 7: Comparison of different density estimation frameworks for a Gaussian mixture model. First line: the estimated densities. Second line: the corresponding transport map. Note, that we do not have access to the analytic optimal transport map in this example.

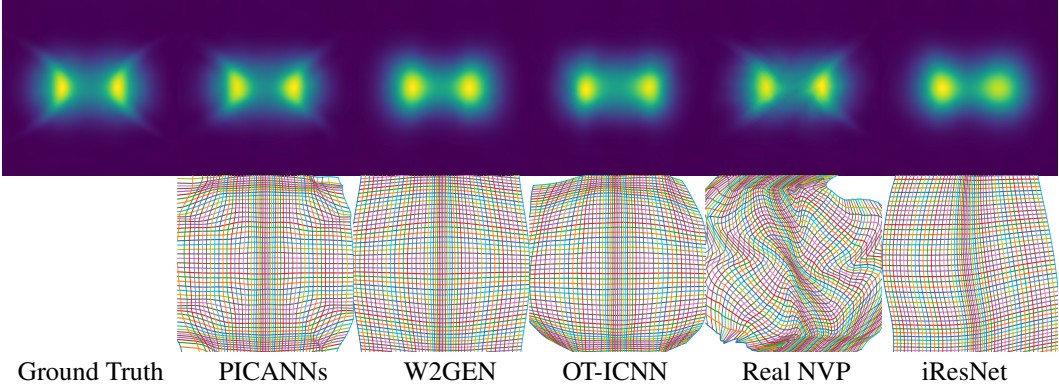

Ground Truth     PICANNs     W2GEN     OT-ICNN     Real NVP     iResNet

Figure 8: Comparison of different density estimation frameworks. First line: the estimated densities. Second line: the corresponding transport map. Note, that we do not have access to the analytic optimal transport map in this example.

## D   ERROR BETWEEN TRUE AND APPROXIMATED WASSERSTEIN DISTANCE

In this appendix we present the percentage error between the theoretical and the approximated Wasserstein metric for three different algorithms: PICANNs, OTICNN from Makkuva et al. (2020) and W2Gen from Korotin et al. (2019). The exact experimental setup and the network details are described in section 3.2.

| Avg % error between true and approximated Wasserstein distance | | | | | | | |
|---|---|---|---|---|---|---|---|
| Random Cvx Function | PICANNs | 0.12 | **0.05** | **0.03** | **0.03** | **0.02** | **0.04** |
| | OT-ICNN | 0.10 | 0.10 | 0.08 | 0.07 | 0.10 | 0.09 |
| | W2GEN | **0.09** | 0.067 | **0.03** | 0.04 | 0.06 | 0.52 |
| Random Gaussian | PICANNs | **1.56** | 0.88 | **0.35** | **0.21** | **0.19** | **0.15** |
| | OT-ICNN | 1.66 | 1.40 | 0.93 | 0.95 | 0.27 | 0.31 |
| | W2GEN | 1.59 | **0.75** | 0.41 | 0.25 | 0.35 | 0.19 |
| Annulus | PICANNs | **5.37** | **1.25** | **1.81** | **1.56** | 7.44 | 5.07 |
| | OT-ICNN | 6.54 | 25.89 | 33.50 | 20.88 | 2.03 | 36.13 |
| | W2GEN | 12.36 | 19.39 | 8.10 | 3.34 | **0.96** | **0.66** |

Table 3: We present a comparison between our PICANN approach and Makkuva et al. (2020) and Korotin et al. (2019). In this table, we present the L2-UVP and the percentage error between the theoretical W2 metric and the approximated W2 metric for three experiments. In all these experiments, the source density is the unit Gaussian. The target density in the case of the "Random Cvx Function" experiment is the unit Gaussian deformed by the gradient of a random convex function. In the case "Random Gaussian", the target density is another Gaussian with a randomly sampled mean and co-variance matrix. In the third experiment, the target density is the annulus distribution. The results in the first two experiments are averages of 20 realizations.

