# OpenReview forum: "Physics Informed Convex Artificial Neural Networks (PICANNs) for Optimal Transport based Density Estimation"
_ICLR.cc/2022/Conference — ICLR 2022 Submitted_

### Official Review · Reviewer_ecZV · 2021-11-01

**Correctness:** 3
**Technical Novelty And Significance:** 2
**Empirical Novelty And Significance:** 2
**Recommendation:** 5
**Confidence:** 4

**Main Review:**

The paper is in general well-written and the idea is clearly presented. Yet, I have a few concerns as listed below:

1. The authors stated that "We will, however, see, that in our situation, i.e., for absolutely continuous measures, the existence and uniqueness is indeed guaranteed". This refers to the Brenier's theorem concerning that a unique optimal transport map exists which is given by the gradient of a convex function. This consequently leads to the Monge-Ampere equation that help formulate the loss function in the proposed work. Yet, the physics informed neural networks adopted can only be shown to be a convex function, but there is no guarantee that the optimized network is indeed the solution of the Monge-Ampere equation. Hence, the uniqueness of the map does not seem to be guaranteed. If the alternative is true, a proof is needed here.

2. The presentation of the density estimation in Section 2.3 is a bit confusing. What does it mean to be "or equivalently $\bigtriangledown u)_* \mu = \nu$"? A bigger question is whether the proposed work can be used in density estimation, given that the uniqueness of the mapping is not guaranteed as in Comment 1 above.

3. Although literature review covers other density estimation techniques such as normalizing flow, there is no comparison with any of these algorithms in Section 3.3. The authors stated in Section 1 that "Although these methods (normalizing flow) have shown to perform well in density estimation applications, the interpretability of the obtained transformation is less clear". But it is not clear to me how the interpretability of the proposed method is clearer, especially given the fact that the computed density may not be exact/correct (due to the lack of the proof of the uniqueness of the obtained approximate mapping), while normalizing flow admits exact density estimation.

**Summary Of The Paper:**

The paper presented an input convex neural networks (ICNN)-based methods to approximate the optimal transport maps. The learning objective is based on the Monge-Ampere equation, i.e. zero loss corresponds to the solution to the Monge-Ampere equation. Physics Informed Neural Networks (PINNs) are adopted to minimise the loss. The author(s) also extended the work to density estimation. The problem is interesting as estimation of optimal maps has a wide range of applications. Experiment results look promising when compared with other PINNs in one experiment setup, but only qualitative results are provided for the density estimation experiment with no comparison of other baselines.

**Summary Of The Review:**

The paper provides an interesting approach to incorporate ICNN and Brenier's Theorem to approximate optimal transport between distributions. Experiment results are encouraging. But there is no proof to support the claim on the uniqueness of the mapping. Therefore, its application on density estimation is also not fully supported.

---

> ### Author Response · Authors · 2021-11-16
> **Response to Reviewer ecZV**
>
> We thank the referee for his/her comments. Below we will provide detailed answers to his/her main remarks.
>
> >This refers to the Brenier's theorem concerning that a unique optimal transport map exists which is given by the gradient of a convex function. This consequently leads to the Monge-Ampere equation that help formulate the loss function in the proposed work. Yet, the physics informed neural networks adopted can only be shown to be a convex function, but there is no guarantee that the optimized network is indeed the solution of the Monge-Ampere equation. Hence, the uniqueness of the map does not seem to be guaranteed. If the alternative is true, a proof is needed here.
>
> We respectfully disagree with this comment. We agree that the ICNN structure only guarantees the convexity of the obtained solution map. The Monge-Ampere equation is, however, encoded in our approach by using it as loss function in the optimization procedure. Consequently, our optimization procedure searches for the optimal approximation of the true solution to the Monge-Ampere equation for a given ICNN structure. In spirit, this approach of solving a PDE using a neural network, does not differ fundamentally from more traditional methods of numerical analysis, e.g. a finite element methods (FEM) approach, where the PDE is similarly solved in a weak formulation. The main difference is the solution space; instead of the space spanned by the FEM basis, in the PINN approach the solution space is spanned by the neural network. By the universal approximation property of neural networks this implies that we can approximate the true solution arbitrarily well by increasing the complexity of the used network. In addition we have shown in our experiments that, already for a relatively simple network structure, the loss function takes values close to zero, cf. the results in Table 1.
>
> Finally, we want to emphasize, that this approach of solving PDEs, by using the weak formulation as loss function for a neural network, is not a new idea from our group, but has has been pioneered by Karniadakis and many others over the past years and has been proven successful in a variety of applications. Similar as in the case of more traditional methods (such as finite element methods) convergence results and error estimates have been obtained, see e.g. [1]
>
> [1] Y. Shin, J. Darbon, G. E. Karniadakis, On the convergence of physics informed neural networks for linear second-order elliptic and parabolic type PDEs, Commun. Comput. Phys., 28, pp. 2042-2074 (2020).
>
> > What does it mean to be "or equivalently $(\nabla u)_* \mu=\nu$?
>
> The density matching constraint can be expressed either via the pull back or via the push forward operation. We have made this more clear in the new version and refer back to the section where these operations are formally introduced.
>
> >But it is not clear to me how the interpretability of the proposed method is clearer, especially given the fact that the computed density may not be exact/correct (due to the lack of the proof of the uniqueness of the obtained approximate mapping), while normalizing flow admits exact density estimation.
>
> We hope that our above answer regarding the uniqueness of the map provides an answer to the uniqueness part of this comment. We want to emphasize here that (theoretically) both normalizing flows and OMT based methods lead to exact density registration algorithms, the difference is merely in the solution selection. While the OMT solution is based on minimizing a physically motivated energy, the solution selection in normalized flows is rather based on computational ease. In the new version we have added practical comparisons to these methods, see also the answer to the last comment below.
>
> As a summary, we believe that an OMT based approach is better interpretable than e.g. normal flows, as the solution is constructed as minimizing a physically motivated loss (distance) function, i.e., the solution of the OMT problem is chosen as the minimizer of an energy among all transformations that satisfy the required density matching constraint.
>
> >Although literature review covers other density estimation techniques such as normalizing flow, there is no comparison with any of these algorithms in Section 3.3.
>
> In this article we focused on the comparison to other OMT solvers, as our main motivation for the present article was the development of a new deep learning based OMT solver. As requested by the referee we now add a comparison to normalizing flow methods, and in particular to RealNVP and iResNet, see Section~3.2 and Appendix B.

---

> > ### Comment · Reviewer_ecZV · 2021-11-18
> > **Uniqueness**
> >
> > Thanks the authors for posting the response. My concern on the uniqueness of the map remains after reading the response. The idea that Monge-Ampere equation is used to construct a loss function does not guarantee that zero-loss will be achieved, which indicates that the obtained solution will be, as the authors mentioned, an "approximation of the true solution". It is hence not theoretically justified to use properties of solutions to Monge-Ampere equation to claim without a proof about the properties of the proposed solution, e.g. "We will, however, see, that in our situation, i.e., for absolutely continuous measures, the existence and uniqueness is indeed guaranteed". As I wrote in the initial review, "the uniqueness of the map does not seem to be guaranteed. If the alternative is true, a proof is needed here."

---

> > > ### Author Response · Authors · 2021-11-19
> > > **Further Clarification about Reviewer ecZV's concerns. (1/2)**
> > >
> > > We thank the referee for taking the time to read our reply and for his/her fast response.
> > >
> > > > It is hence not theoretically justified to use properties of solutions to Monge-Ampere equation to claim without a proof about the properties of the proposed solution, e.g. "We will, however, see, that in our situation, i.e., for absolutely continuous measures, the existence and uniqueness is indeed guaranteed".
> > >
> > > The above quoted sentence is stated in the theoretical background section, and not in the section about our implementation and for the theoretical problem this is exactly the statement of Brenier's theorem. We argue, however, that we never use that our solution is unique and that we also never use any uniqueness properties for any of the applications afterwards. Our line of argument is as follows:
> > > 1) By Brenier's theorem there exists a unique solution to the OMT problem and this unique solution is given by the solution to the Monge-Ampere equation (so far this is not related to our PINNs approach).
> > > 2) We use the PINN approach to approximate this solution, by approximately solving the Monge-Ampere equation. We never claim that this approximation is  unique or exact. In fact in the generic situation, the true solution will not be contained in the space that is spanned by a fixed network and certainly different network sizes will lead to different approximations. In general increasing the network size will led to better and better approximations. This is, however, nothing that is an unique property of our approach, but is true for any numeric discretization. In no part of the article we use any uniqueness properties for this solution. On the contrary, we even state that the quality of our results (and also of the results that are obtained with the other algorithms) really depends on the chosen network structure, see e.g. Page 6, the paragraph that starts with Approximating the OMT map.
> > > 3) We show, however, in experiments that we indeed obtain a good approximation of the true OMT-map, see the results in Table 1, which compares the true OMT maps to the estimated OMT maps. One can clearly see that our approximation outperforms the approximations of two other state-of-the-art deep learning based algorithms for estimating OMT.
> > >
> > > > ...the computed density may not be exact/correct (due to the lack of the proof of the uniqueness of the obtained approximate mapping), while normalizing flow admits exact density estimation.
> > >
> > > We have to admit that we are somewhat confused, what the referee means with exact density estimation. Density estimation is the problem of estimating a density based on a limited number of samples. Thus the notion of an exact density estimation is not well-defined. One could consider the related problem of density registration, where indeed exact solutions exist, i.e., transformations that exactly push the source density to the target density. By Brienner's theorem, the gradient of a convex function is a OMT map between the source density and the transported density. Because of the ICNN constraints the function is always convex and thus the estimated map is always an OMT map. As the referee pointed out, the loss will be not exactly zero, and thus our transported density will not match exactly the target density in density registration. We want to emphasize that this is not an issue for density estimation as there exists no exact target density! In the OMT-comparison experiments (which is essentially density registration) we compare the quality of our approximations to true solutions and indeed the approximations obtained with our algorithm outperform the other two deep-learning based OMT solvers (in terms of accuracy of the OMT map estimation and consequently in terms of the obtained mismatch error). We want to emphasize, however, that normalizing flows also only approximately solve the density registration problem. Indeed, in some of our density estimation experiments, our finite samples were drawn from given ground truth densities, which allows one to estimate the quality of the results by checking how well one can recover this ground truth. In these experiments we have compared our results to results that were obtained with methods of the family of normalizing flows (RealNVP and iResNet) and one can clearly see that the estimated densities are more "exact" with our method as compared to these method (Figures 1,2,7,8, the only case where our method didn't produce the best result is the example in Figure 7, where we are slightly outperformed by iResNet.)

---

> > > > ### Author Response · Authors · 2021-11-19
> > > > **Further Clarification about Reviewer ecZV's concerns. (2/2)**
> > > >
> > > > As a summary: after carefully checking the manuscript again, we didnt find any point in the article where we claim that our approximation is unique, neither do we claim that we have an exact solution to the density registration problem (the relevant experiments section is even called approximating the OMT map). In case, that our formulation regarding this fact is at points unclear, we would ask the referee to point us to the exact places in the paper and we will be happy to clarify any such statement. Furthermore, we want to emphasize again that our experiments clearly show that our results outperform other deep learning OMT solves and also methods from the family of normalizing flows; both in terms of accuracy of the OMT transformation (as compared to the other OMT solvers), but also in terms of the exactness of the obtained density registration (as compared to the other OMT solvers and the two methods from the family of normalizing flows).

---

> > > > > ### Comment · Reviewer_ecZV · 2021-11-28
> > > > > **Thanks for the clarification**
> > > > >
> > > > > In my initial review, I raised that "Hence, the uniqueness of the map does not seem to be guaranteed. If the alternative is true, a proof is needed here." The authors could have directly answered that the uniqueness of the map is not guaranteed in their initial response. Instead, the initial response was focused on stating that such approaches have been adopted by other groups. Only after my follow-up comment, the authors claimed that all statements on the uniqueness is in the theoretical background section and they never use that their solution is unique. The authors could have explicitly addressed this in the first response and can also choose to make it clear in the paper to avoid any such confusion.
> > > > >
> > > > > Regarding the author's confusion "what the referee means with exact density estimation", the authors stated that "the notion of an exact density estimation is not well-defined". Yet, the authors claimed again "Furthermore, we want to emphasize again that our experiments clearly show that our results outperform other deep learning OMT solves and also methods from the family of normalizing flows; both in terms of accuracy of the OMT transformation (as compared to the other OMT solvers), but also in terms of the exactness of the obtained density registration (as compared to the other OMT solvers and the two methods from the family of normalizing flows)." So, it is indeed confusing to me what the authors meant by "exactness" here, as the authors have clarified that this is an approximate solution for OMT. And for normalizing flow, the pushed measure can be evaluated exactly due to the invertibility of the mapping which is not the case here due to the approximate solution to the Monge Ampere equation.

---

> > > > > > ### Author Response · Authors · 2021-11-29
> > > > > > **Clarification on invertibility of approximated maps**
> > > > > >
> > > > > > We thank the referee for his responsiveness and we are glad that our second answer was able to clarify the situation. It seems, however, that there is still some confusion regarding the invertibility of our estimated maps. Thus we want to emphasize that the invertibility of the transport maps is entirely unrelated to the Monge-Ampere equation, but stems solely from the fact that we represent our map as the gradient of a strictly convex function, which is guaranteed by the ICNN structure (the Jacobian of the map is the hessian of the potential function, which is thus symmetric and positive definite as the function is strictly convex). Thus all the maps, that are obtained with our approach are necessarily invertible and thus, exactly as for the normal flow methods, the pushed measure can be evaluated exactly. The Monge-Ampere equation is used to enforce the matching constraint, i.e., to enforce that the estimated transformation really transports the background density to the required target density (or in the situation where we only have a finite number of sample from an unknown target density, that samples from the pushed density match this discrete samples in the KL-divergence). Finally, we invite the referee to inspect the figures 1,2,7,8 which (as we believe) clearly demonstrate what we mean by "more exact" as compared to methods from normal flows.

---

### Official Review · Reviewer_U76A · 2021-11-02

**Correctness:** 3
**Technical Novelty And Significance:** 4
**Empirical Novelty And Significance:** 3
**Recommendation:** 8
**Confidence:** 3

**Main Review:**

The paper combines a range of interesting ideas to solve density estimation problems. The authors reformulate the original OMP problem as a special PDE. In turn, PINNs can be used to approximate the solution of this special PDE. In addition, the authors use input convex neural networks (ICNNs) to  incorporate the Brenier potential into the PINN. This approach bypasses some of the challenges of directly minimizing the original OMP problem.

* _Originality:_ The presented ideas in this paper are original and well suited for this conference. I not aware of another paper that uses PINNs for density estimation.

* _Quality:_ The proposed framework seems plausible and is demonstrated on a few experiments. There is no theory that backups some of the claims, and the amount of experiments is on the lower end of the spectrum for an empirical ML conference paper. Further, it would help to better discuss the assumption that have been made and some of the limitations. I assume that the authors have a fair understanding for these details and they should make better use of the unlimited space of the appendix to provide additional discussions and experiments.

* _Clarity:_ The proposed ideas are well organized and clearly presented. The detailed discussion of the background materials helped reading the paper.

* _Significance:_ This paper is of interest for two communities within ML. First, it is of interest for the PINN community and the paper has the potential to motivate future research in the intersection of PINNs and density estimation. On the other hand, density estimation is an important topic in ML and the presented approach performs well on the presented tasks.However, it is not clear whether the presented approach is competitive with other state-of-the-art methods in this area, or whether it just does better on the selected tasks compared to 2 baselines. A better discussion of the advantages and limitations and additional baselines would help to strengthen the paper and improve its significance.


Here are a few comments that might help to improve the paper:

* The weakness of this paper is the experimental section. Only two other algorithms are used for comparison. The authors state that these two algorithms performed best on their benchmark problems, still it would be interesting to show results for other algorithms that have been tested. Also, how does your approach compare to state-of-the-art GANs or neural ordinary differential equations (NODEs) for density estimation methods? At least, it would be helpful to discuss some qualitative advantages / disadvantages of your proposed approach in comparison to other methods such as GANs or NODEs. In which situation should one use your approach, and which situation does your approach fail?

* You state that you only consider the simple case of a quadratic cost function. Does Brenier's Theorem apply to other cost functions?

* You state that ICNNs introduce implicit regularization. Can you provide theory to back this claim up, or is this only an educated guess?

* The results in Figure 6 look interesting. Why don't you include these results in the main text? I would rather move the network details to the appendix.

* Can you comment on the computational costs of your framework as compared to the other methods that you used for comparison.

* On page 2: I assume that there is some typo in:  'the convex function itself should be indenity helps in avoiding'.

* On page 9: word repetition in 'a density estimation estimation'.

* The quality of the figures is poor. It would be nice to better format the figures so that they eat up less space and use vector graphics instead of low-resolution pngs.

**Summary Of The Paper:**

This paper introduces a new method for density estimation that is based on the idea of optimal mass transport (OMP), i.e., finding some function that transports a probability density into another probability density while minimizing the transportation cost. To do so, the authors leverage Brenier's theorem which guarantees the existence and uniqueness of the optimal transport map. As a consequence of this theorem,  the optimal transport map of the OMT problem can be obtained as the solution of a special nonlinear partial differential equation (PDE). The authors then show that the approximate solution can be computed with help of physics informed neural networks (PINNs). The approach is demonstrated for several canonical examples.

**Summary Of The Review:**

This paper presents interesting and novel ideas for density estimation, and to the best of my knowledge this is the first paper that uses PINNs for density estimation. The paper is clearly written and provides a good overview of related work and background to put the proposed ideas into context. However, since this paper provides only empirical results and no theory, I only think that the paper is slightly above the acceptance threshold. Additional results would help to make the story more compelling, and given additional results I am happy to reconsider my rating.

---

> ### Author Response · Authors · 2021-11-16
> **Response to Reviewer U76A**
>
> We thank the referee for his/her comments. Below we will provide detailed answers to his/her main remarks.
>
> >The weakness of this paper is the experimental section. Only two other algorithms are used for comparison. The authors state that these two algorithms performed best on their benchmark problems, still it would be interesting to show results for other algorithms that have been tested. Also, how does your approach compare to state-of-the-art GANs or neural ordinary differential equations (NODEs) for density estimation methods? At least, it would be helpful to discuss some qualitative advantages / disadvantages of your proposed approach in comparison to other methods such as GANs or NODEs. In which situation should one use your approach, and which situation does your approach fail?
>
> The main motivation of the present article is the development of a new deep learning based OMT solver, which is of relevance in several fields beyond density estimation. Thus we focused our experiments mainly on comparisons with other OMT solvers and on the approximation error of our solutions as compared to true (analytic) OMT solutions. Therefore we chose the best performing algorithms based on a recent benchmarking paper as a basis for our comparisons. As suggested by the referee we have now added comparisons to other density estimation techniques from the family of normalized flows: RealNVP and iResNet, see Section 3.2 and appendix C. We want, however,  to emphasize that although NODE and normalized flows have been proposed for density estimation they do not solve the OMT problem and consequently there is no physically motivated optimality conditions of the transport map generated by these methods. This is also reflected in our experiments: in the OMT approach we search for the map that is as close as possible to the identity among all transformations that satisfy the required density matching constraint (note, that there is an infinite dimensional set of transformations that satisfy this constraint). Consequently the obtained transformation maps for OMT are more regular as compared to the maps obtained with other density estimation methods such as iResNet or RealNVP, cf. Figures 1,2,7 and 8. Thus we believe that, in general, the OMT framework provides a theoretically and practically more sound framework. We have added a comment on this on Page 8.
>
> >You state that you only consider the simple case of a quadratic cost function. Does Brenier's Theorem apply to other cost functions?
>
> The reason for this restriction is precisely, that Brenier's theorem only holds in this case. We have changed the formulation and clearly state this now.
>
> >You state that ICNNs introduce implicit regularization. Can you provide theory to back this claim up, or is this only an educated guess?
>
> The implicit regularization properties of ICNNs have been recently proven in the article [1].
> We have added this reference on Page 5, last paragraph.
>
> [1] Sivaprasad S., Singh A., Manwani N., Gandhi V. (2021) The Curious Case of Convex Neural Networks. In: Oliver N., Pérez-Cruz F., Kramer S., Read J., Lozano J.A. (eds) Machine Learning and Knowledge Discovery in Databases. Research Track. ECML PKDD 2021. Lecture Notes in Computer Science, vol 12975. Springer, Cham. https://doi.org/10.1007/978-3-030-86486-6_45
>
> > The results in Figure 6 look interesting. Why don't you include these results in the main text? I would rather move the network details to the appendix.
>
> We thank the reviewer for this suggestion, which we happily followed. In addition we have added comparisons to a similar setup using 4 other density estimation techniques.
>
> >Can you comment on the computational costs of your framework as compared to the other methods that you used for comparison.
>
> As remarked by reviewer 44BN the presence of the determinant in our loss functions leads to exponential scaling with respect to dimensions of the domain of integration Omega, which is certainly a disadvantage of the proposed method. We believe that approximating the determinant in higher dimensions could help to alleviate this fact. We have added remarks on this fact and also comment on the comparison to other methods. In addition we present now the convergence plots for selected experiments and present timings for the different articles, see Appendix B.
>
> >On page 2: I assume that there is some typo in: 'the convex function itself should be identity helps in avoiding'.?
>
> Yes, this whole sentence was a typo and has been removed.
>
> >On page 9: word repetition in 'a density estimation estimation'.
>
> We corrected this mistake.
>
> >The quality of the figures is poor. It would be nice to better format the figures so that they eat up less space and use vector graphics instead of low-resolution pngs.
>
> We now use vector graphics for all figures.

---

> > ### Comment · Reviewer_U76A · 2021-11-19
> > **Thank your for the response**
> >
> > The authors addressed all my concerns and I am happy to raise the score for this paper.
> >
> > Overall, I think that this paper is an interesting contribution for this conference, despite some of its weaknesses. I am in favor of this paper, because it combines a range of interesting ideas and connects machine learning with ideas from dynamical systems. I couldn't see that any of the other reviewers raised a strong point that would let me doubt my score. Further, the discussion revealed that there are some interesting follow-up questions that motivate future work, e.g.,  studying whether efficient determinant estimators help to scale the proposed method.
> >
> > I agree with other reviewers that the experiments are on the weaker side for a conference paper at ICLR. But, the authors have started to improve the experiments during the rebuttal phase. I would like to ask the reviewers to continue to work on the experiments for the camera ready version (conditional that the paper gets accepted). In general, it is nice to see more experiments, in particular since this is `a new deep learning based OMT solver' and since there are no strong theoretical results. Given extra time, I am confident that the reviewers can expand the result section and further strengthen the paper.
> >
> > As a final comment, I don't feel strongly about that this method provides only an approximation for the OMT map. At the very least the experiments indicate that the model learns a 'useful' approximation. To quote John Tukey: "An approximate answer to the right problem is worth a good deal more than an exact answer to an approximate problem."

---

### Official Review · Reviewer_44BN · 2021-11-03

**Correctness:** 3
**Technical Novelty And Significance:** 2
**Empirical Novelty And Significance:** 2
**Recommendation:** 6
**Confidence:** 5

**Main Review:**

Strength: + The probabilistic universal approximation framework can be much more efficient than the finite element or difference methods which is known to have exponential scaling with respect to dimensions.
+ The paper provides sufficient details in terms of network, hyperparameters, evaluation methods in the experimental section.

Weakness: - The main required definitions are provided, however, the main relevant technical concepts are explained heuristically. For example, while relevant background for optimal transport theory is provided in detail, there is little to no background on partial differential equations.
- The main loss functions (4) and (5) proposed do have exponential scaling with respect to dimensions of the domain of integration \Omega.
- Density estimation experiments are preliminary.


**Summary Of The Paper:**

The paper proposes a new physics informed neural architecture for efficiently learning and optimizing optimal transport maps using techniques from partial differential equations.

**Summary Of The Review:**

Justification: The paper makes a solid contribution in terms of technical content. The main high level idea is that solving optimal transport maps under some conditions can be posed as a PDE. While this is not something new, for instance, sinkhorn algorithm does the same -- turn a nonsmooth optimization problem, a linear program, into a smooth version, as given in equations (4) and (5). However, many technical details regarding the algorithm are missing -- (i) what is the overall procedure?; (ii) near equation (7), there is some discussion regarding representing u again using a neural network, is the overall architecture different from what is shown in Figure (1)? It seems like some of these questions regarding how and what PDE is exactly being solved can be answered by reading various other papers cited in the experiments, but it is not clear from the submission. In this sense, I feel like the paper spends too much time on background material (which the reader can pick up from any standard book on Applied PDE, for example, Introduction in Optimal
Transport for Applied Mathematicians by Filippo Santambrogio, 2015) rather than focusing on the key insights of the paper.

The network details are provided in abundant details for verification of the main idea. However, the main task density estimation which the paper mentions quite in detail about in the beginning of the paper is only minimally experimented. For example, the paper claims that finite elements solvers are slow in high dimensions. However, the paper contains no comparisons with these methods on the problems considered in the paper, which are also low dimensional.

---

> ### Author Response · Authors · 2021-11-16
> **Response to Reviewer 44BN**
>
> We thank the referee for his/her comments. Below we will provide detailed answers to his/her main remarks.
>
> >The main loss functions (4) and (5) proposed do have exponential scaling with respect to dimensions of the domain of integration Omega.
>
> We thank the referee for this excellent comment and agree that this is indeed a drawback of our method as presented. This stems from the fact that the Monge-Ampere equation contains the determinant of the Jacobian. Using the positive definiteness of the Jacobian (being the Hessian of a convex function) one could potentially make use of efficient determinant estimators, such as Stochastic Chebyshev Expansions as developed in [1].
>
> The focus of our present work was to obtain as precise as possible solution to the OMT problem and thus we refrained from doing so. We have added a comment on the exponential scaling and how we could potentially overcome it (Page 5) and we plan to follow this line of research in future work.
>
> [1] Han, Insu, Haim Avron, and Jinwoo Shin. "Stochastic Chebyshev Gradient Descent for Spectral Optimization." 32nd Conference on Neural Information Processing Systems (NIPS).
> Neural Information Processing Systems Foundation, 2018.
>
> > Density estimation experiments are preliminary.
>
> The main motivation of the present article is the development of a new deep learning based OMT solver. Thus, in the previous version, we focused our experiments mainly on comparisons with other OMT solvers and on the approximation error of our solutions as compared to true (analytic) OMT solutions.  In the new version we have now added comparisons to further density estimation methods, namely (W2GEN, OT-ICCNN, RealNVP and iResNet); see Fig. 1 and the examples in Appendix. In addition, while we are at the current time not able to present comprehensive  experiments for the generative power of the obtained framework we moved, as suggested by Reviewer U76A, the MNIST example from the appendix to the main part of the article (Section 3.3), where we also compare the generative model to a similar setup using four other density estimation frameworks.
> We hope that these changes help with improving this aspect of our article.
>
> > near equation (7), there is some discussion regarding representing u again using a neural network, is the overall architecture different from what is shown in Figure (1)?
>
> We indeed use the same architecture. We added a comment on this fact.
>
> > what is the overall procedure? ... It seems like some of these questions regarding how and what PDE is exactly being solved can be answered by reading various other papers cited in the experiments, but it is not clear from the submission. In this sense, I feel like the paper spends too much time on background material (which the reader can pick up from any standard book on Applied PDE, for example, Introduction in Optimal Transport for Applied Mathematicians by Filippo Santambrogio, 2015) rather than focusing on the key insights of the paper.
>
> We really feel that the background on OMT is crucial for this paper, and so we are hesitant to further shorten this section. We have, however, compactified some of the formulas, which allowed us to spend only one page on this section in the new version. We have used this extra space to describe the proposed method for solving the PDE in more details. In particular we have added a new paragraph around equation (3), that outlines the general procedure and specifies the exact PDE. In addition we have added a paragraph on the solution theory of the Monge-Ampere equation, see Page 4. We hope these changes make this part of the article better understandable.

---

### Official Review · Reviewer_U1ba · 2021-11-05

**Correctness:** 4
**Technical Novelty And Significance:** 4
**Empirical Novelty And Significance:** 1
**Recommendation:** 5
**Confidence:** 3

**Main Review:**

The proposed method in this paper has the potential to be a very useful method for solving the continuous optimal transport problem using DNNs.
In addition, the manuscript is clear and well constructed.
On the other hand, the effectiveness of the method for application problems in machine learning cannot be confirmed because there are few implications about real applications and few experiments.
In order for the paper to be accepted at this conference, it might be necessary to apply the method to a real problem such as the image generation problem of natural images, and then conduct experiments to compare the method with existing methods which solved the application problem.
In addition to that, the following points were unclear and I would like to ask for your response.

Does ICNN have the ability to represent all convex functions?

On page5, it says "L^2-norm used in equation 5", but isn't it a mistake of "equation 4"?

In the experiment of method comparison, is there any evidence that the parameters of DNN in the previous study are optimized? (If there is no evidence, it would be better to specify the possibility that it is not optimized?　That would be positively evaluated as a scientific paper, as it would present the limits of verification).

**Summary Of The Paper:**

This paper proposes to solve the continuous Optimal Mass Transport problem as a convex mapping function estimation problem based on Brenier's theorem using input convex neural networks.
The proposed method is compared with other DNN-based methods in the ubiquitous density estimation and generative modeling tasks in statistics, and good results are obtained.

**Summary Of The Review:**

The proposed method in this paper has the potential to be a very useful method for solving the continuous optimal transport problem using DNNs.
In addition, the manuscript is clear and well constructed.
On the other hand, the effectiveness of the method for application problems in machine learning cannot be confirmed because there are few implications about real applications and few experiments.
In order for the paper to be accepted at this conference, it might be necessary to apply the method to a real problem such as the image generation problem of natural images, and then conduct experiments to compare the method with existing methods which solved the application problem.

---

> ### Author Response · Authors · 2021-11-16
> **Response to Reviewer U1ba**
>
> We thank the referee for his/her comments. Below we will provide detailed answers to his/her main remarks.
>
> >Does ICNN have the ability to represent all convex functions?
>
> Yes, ICNNs inherit the universal approximation properties of standard feed forward networks. We added a remark on this fact and provide a reference (Page 4, Section 2.2 paragrph 2).
>
> >On page5, it says "L^2-norm used in equation 5", but isn't it a mistake of "equation 4"?
>
> We thank the referee for pointing us towards this error, which we have corrected in the new version.
>
> >In the experiment of method comparison, is there any evidence that the parameters of DNN in the previous study are optimized? (If there is no evidence, it would be better to specify the possibility that it is not optimized?　That would be positively evaluated as a scientific paper, as it would present the limits of verification).
>
> With exception of some experiments for the OT-ICNN solver all our experiments showed a good convergence behavior. We added a comment on this on Page 6, Section 3.2 Paragraph 1 and provide selected convergence graphs in the appendix.
>
> >On the other hand, the effectiveness of the method for application problems in machine learning cannot be confirmed because there are few implications about real applications and few experiments. In order for the paper to be accepted at this conference, it might be necessary to apply the method to a real problem such as the image generation problem of natural images, and then conduct experiments to compare the method with existing methods which solved the application problem.
>
> The main focus of the current paper was the development of a new deep learning based solver for OMT which, as we believe, will have wide applications not only in machine learning, but also in areas such as medical imaging or shape analysis. We agree that a comparative image generation experiment would be very interesting.  While we are unfortunately not able to present such a large scale experiment at the current time, we plan to follow this line of research in future work. In the revised version we present, however, further comparisons to other density estimation algorithms, such as normalized flows, see Section 3.2. These new experiments further validate the performance of our proposed algorithm. In addition, as suggested by Reviewer U76A, we moved the generative MNIST example from the appendix to the main part of the article. In addition we compared the generative model to a similar setup using four other density estimation frameworks. We hope that these new experiments and changes help with improving this aspect of our article.

---

### Author Response · Authors · 2021-11-16
**Updates following the reviewers’ comments.**

We thank all the reviewers for their feedback, and detailed comments. We have incorporated most of the suggestions of the reviewers. We describe here the overall updates in our new revision. Responses to individual comments are posted as replies to the individual reviews:

Section 1: With the exception of the correction of a typo and an update in the contributions part this section is essentially unchanged.

Section 2.1: The reason for restricting to quadratic cost functions has been clarified and a comment on the solution theory for the Monge-Ampere equation has been added

Section 2.2: A comment on the universal approximation property of ICNNs and a paragraph to detail our general procedure have been added.

Section 2.3: minor typos have been corrected and a reference for the implicit regularization properties of ICNNs has been added.

Section 3.1: The figure describing the network architecture for our PICANNs approach has been moved to the appendix.

Section 3.2 and 3.3 have been merged and several new comparison experiments have been added. In particular comparisons to two new algorithm have been added (RealNVP and iResNet). To make space for these new experiments a part of table 1 and the gaussian mixture figure have been moved to the appendix.

Section 3.3 was previously in the appendix and has now been included in the main part of the paper. In addition we compared the generative model to a similar setup using four other density estimation frameworks.

Appendix B, which shows several convergence plots and timing results for the three OMT algorithms, has been added

Appendix C, which shows further density estimation experiments has been added.

Appendix D, which contains the average error between the true and the approximated Wasserstein distance for the experiments in Section 3.2 has been added.

---

### Decision · Program_Chairs · 2022-01-20

**Decision:**

Reject

**Comment:**

This work proposes to define densities via the pushforward of a base density through the gradient field of a convex potential as studied in OT theory and, in particular, inspired by Brenier's theorem.

More concretely, it proposes to use ICNNs to parametrize the convex potentials and considers two mechanisms to match a target density: 1) with a known (normalized) target approximately solve the Monge-Ampere equation via optimization; 2) with only samples available, they propose to use the maximum-likelihood approach.

While the paper is overall well-written, the idea is very close to existing work that was not mentioned or discussed in the paper. The paper would benefit from a substantial revision to incorporate the missing references and emphasize the relative novelty.